# Hypertension in African Populations: Review and Computational Insights

**DOI:** 10.3390/genes12040532

**Published:** 2021-04-06

**Authors:** Sihle E. Mabhida, Lebohang Mashatola, Mandeep Kaur, Jyoti R. Sharma, Teke Apalata, Babu Muhamed, Mongi Benjeddou, Rabia Johnson

**Affiliations:** 1Biomedical Research and Innovation Platform, South African Medical Research Council, Tygerberg 7505, South Africa; sihlemabhida@gmail.com (S.E.M.); jyoti.sharma@mrc.ac.za (J.R.S.); 2Department of Biotechnology, Faculty of Natural Science, University of the Western Cape, Private Bag X17, Bellville, Cape Town 7535, South Africa; mbenjeddo@uwc.ac.za; 3School of Molecular and Cell Biology, University of the Witwatersrand, Private Bag 3, Johannesburg 2050, South Africa; 681452@students.wits.ac.za (L.M.); mandeep.kaur@wits.ac.za (M.K.); 4Division of Medical Microbiology, Department of Laboratory-Medicine and Pathology, Faculty of Health Sciences, Walter Sisulu University and National Health Laboratory Services, Mthatha 5100, South Africa; tapalata@wsu.ac.za; 5Hatter Institute for Cardiovascular Diseases Research in Africa, Department of Medicine, University of Cape Town, Cape Town 7535, South Africa; babu.muhamed@uct.ac.za; 6Children’s National Health System, Division of Cardiology, Washington, DC 20010, USA; 7Division of Medical Physiology, Faculty of Medicine and Health Sciences, Stellenbosch University, Tygerberg 7505, South Africa

**Keywords:** hypertension, pharmacogenomics, single-nucleotide polymorphism, Africa, genetic variation

## Abstract

Hypertension (HTN) is a persistent public health problem affecting approximately 1.3 billion individuals globally. Treatment-resistant hypertension (TRH) is defined as high blood pressure (BP) in a hypertensive patient that remains above goal despite use of ≥3 antihypertensive agents of different classes including a diuretic. Despite a plethora of treatment options available, only 31.0% of individuals have their HTN controlled. Interindividual genetic variability to drug response might explain this disappointing outcome because of genetic polymorphisms. Additionally, the poor knowledge of pathophysiological mechanisms underlying hypertensive disease and the long-term interaction of antihypertensive drugs with blood pressure control mechanisms further aggravates the problem. Furthermore, in Africa, there is a paucity of pharmacogenomic data on the treatment of resistant hypertension. Therefore, identification of genetic signals having the potential to predict the response of a drug for a given individual in an African population has been the subject of intensive investigation. In this review, we aim to systematically extract and discuss African evidence on the genetic variation, and pharmacogenomics towards the treatment of HTN. Furthermore, in silico methods are utilized to elucidate biological processes that will aid in identifying novel drug targets for the treatment of resistant hypertension in an African population. To provide an expanded view of genetic variants associated with the development of HTN, this study was performed using publicly available databases such as PubMed, Scopus, Web of Science, African Journal Online, PharmGKB searching for relevant papers between 1984 and 2020. A total of 2784 articles were reviewed, and only 42 studies were included following the inclusion criteria. Twenty studies reported associations with HTN and genes such as *AGT* (rs699), *ACE* (rs1799752), *NOS3* (rs1799983), *MTHFR* (rs1801133), *AGTR1* (rs5186), while twenty-two studies did not show any association within the African population. Thereafter, an in silico predictive approach was utilized to identify several genes including *CLCNKB*, *CYPB11B2*, *SH2B2, STK9*, and *TBX5* which may act as potential drug targets because they are involved in pathways known to influence blood pressure. Next, co-expressed genes were identified as they are controlled by the same transcriptional regulatory program and may potentially be more effective as multiple drug targets in the treatment regimens for HTN. Genes belonging to the co-expressed gene cluster, *ACE, AGT, AGTR1, AGTR2*, and *NOS3* as well as *CSK* and *ADRG1* showed enrichment of G-protein-coupled receptor activity, the classical targets of drug discovery, which mediate cellular signaling processes. The latter is of importance, as the targeting of co-regulatory gene clusters will allow for the development of more effective HTN drug targets that could decrease the prevalence of both controlled and TRH.

## 1. Introduction

Hypertension (HTN), also known as high blood pressure (BP), can be classified into primary or secondary HTN with secondary HTN affecting 5–10% of hypertensive patients with the cause linked to an underlying medical condition. In contrast, primary hypertension also known as essential HTN accounts for 90–95% of HTN cases, it has no identifiable secondary root and is the most common cause of stroke and cardiovascular disease (CVD) [1,2]. Accumulative evidence showed that the prevalence of HTN is 1.3 billion, and this number is projected to increase to 1.56 billion by 2030 with an estimated global economic cost of $274 billion [1,2,3,4]. Despite the availability of advanced diagnostic options and the use of multiple antihypertensive medications, many studies have reported inadequate control of blood pressure among hypertensive individuals worldwide, including Africa [2,5,6,7]. This inability to control blood pressure even after adherence has been referred to as treatment-resistant hypertension (TRH). According to the American heart association, TRH is defined as a BP which remains ≥140/90 mg despite the use of three or more antihypertensive drugs [8,9].

Although poor control of high blood pressure may have many causes, the most likely contributor is the poor response to the prescribed antihypertensive drug [10]. Current drugs are used to treat HTN without an in-depth understanding of the biological basis that regulates the disease or the effect of an individual’s make-up on the efficacy of the drug to control HTN [11]. The most commonly prescribed drug classes to treat hypertension include diuretics and vasodilators designed to reduce vascular resistance. The major disadvantages are side effects (i.e., renal dysfunction) and unpredicted blood pressure responses in patients (i.e., HTN) [12,13]. Drug classes developed from the knowledge of biological pathways include the angiotensin II antagonists and angiotensin-converting enzyme inhibitors. These have been developed through the understanding of the renin-angiotensin-aldosterone system (RAAS) in the regulation of blood pressure. As such, first-line therapy includes angiotensin-converting enzyme (*ACE*) inhibitors, angiotensin II receptors, and calcium-channel blockers [14,15]. These have proven to be the most effective drug classes (coupled with lower side effects) when used as a combination therapy [16]. Nonetheless, though effective, the one-drug-fits-all model has been proven to be ineffective due to an individual’s genetic make-up and variable response to several antihypertensive drugs could be attributed to the genetic variability in the candidate BP regulating genes and their pathways [17]. Many studies have demonstrated that genetic factors are responsible not only for blood pressure elevation but also play a profound role in the interindividual variability in drug response, offering an opportunity for pharmacogenomic investigation and potential individualized drug therapy [18,19]. Therefore, it has been speculated that a patient’s inter-individual genetic variation can be used to understand the pathophysiology of the disease and that this pharmacogenomics approach may have the potential of individualizing HTN drug therapy based on the patient’s genetic background [20,21]. For example, Guo et al., [22] and Choi et al., [23] have reported successful examples of genetic polymorphisms on blood pressure response to antihypertensive therapies. Recently, Ma et al. [21] reported that various informative genetic variants can be utilized for the identification of subtypes of hypertension. These studies have highlighted the importance of understanding the biological basis of the disease and the contribution of genetic variations in the development and progression of the disease.

Despite major advances being made over the last decade, understanding the role of genetics governing the phenotypic state of HTN is complex because on one hand there are rare monogenic hypertensive syndromes while on the other side are the cases of primary or TRH, which may occur due to a varied expression and interactions of multiple genes [24]. However, polygenic inheritance patterns in such cases are not only complex but also enigmatic.

Nonetheless, various subtle genetic variations have been identified through approaches such as genome-wide association studies, and these small variations among individuals may account for genetic susceptibility to HTN. However, these single-nucleotide polymorphisms (SNP) have been shown to vary across racial and ethnic groups and play a pivotal role during the development and progression of this phenotypic state [25]. SNPs are changes in specific nucleotides at fixed positions in DNA sequence and occur frequently once in every 1000 base pairs. These are the most commonly occurring genetic variations causing varied inter-individual drug response which remains a major public health concern [1,3,26]. Furthermore, it has been speculated that variable response to drugs could also be due to different subtypes of a disease phenotype in certain individuals. To exemplify, patients demonstrating better BP response with certain drugs may have different pathophysiology of HTN than others showing poor response with those drugs [18]. Therefore, a thorough understanding of the genetic background of HTN is critical to predict an individual’s disease risk as well as to improve individualized treatment response. Despite many SNPs having no clinical significance, several key SNPs have been reported with a detailed analysis through pharmacogenomics-based studies playing an important role in antihypertensive drug response [25,27].

As such, the purpose of this review was to systematically extract and discuss African evidence on the effect of genetic variations in hypertensive patients and use in silico methods to further elucidate biological processes that will aid in identifying novel drug targets for the treatment of HTN in an African population.

## 2. Materials and Methods

A systematic search was performed using subject headings or primary search terms such as resistant hypertension, genetics, SNP, and pharmacogenomic under the Preferred Reporting Items for Systematic Review (PRISMA) guidelines [28]. This was done using major search engines and databases such as PubMed, Scopus, Web of Science, African Journal Online, and PharmGKB were used for the search as shown in Appendix A.

### 2.1. Inclusion Criteria and Data Extraction

The studies were included in the systemic review based on the following criteria: (i) Investigated the association between SNPs and HTN in African-based population, (ii) Used case-control design, (iii) Published from 1984 to 2020, (iv) Studies done on humans (Table 1). Studies were excluded if duplicate publications reported selectively on migrant Africans outside Africa, family studies, used linkage analysis and analyzed mixed populations of African descent without considering their country of residence. Moreover, language restriction was applied with studies conducted in languages other than English were excluded. The articles were independently assessed for compliance with the inclusion or exclusion criteria by two authors resolving disagreements and reached a consistent decision. The following information was extracted from each study: first author and year of publication; country of origin; ethnicity of the study population; the number of subjects under hypertensive cases and controls; diagnostic criteria for hypertensive cases and controls and SNPs analyzed.

### 2.2. Drug–Gene Interaction

The bioinformatics tool iCTNET from Cytoscape (available at http://apps.cytoscape.org/apps/iCTNET (accessed on 10 March 2020) was used to search through the DrugBank (available at https://www.drugbank.ca (accessed on 10 March 2020) and comparative toxicogenomic database (CTD) (available at http://ctdbase.org/ (accessed on 10 March 2020) to identify drug–gene interactions. These were coupled with the occurring frequent and infrequent side-effects for each drug and plotted in a network. This analysis helped to identify the common drug-targets of HTN.

### 2.3. Co-Expression Networks

Networks of co-expressed genes were generated from mathematical models which were used to predict protein–protein interactions (PPIs) using the bioinformatics tool Expression Correlation from Cytoscape [29] (available at http://apps.cytoscape.org/apps/Expression Correlation (accessed on 9 March 2020). The reference database Expression Correlation utilizes sample data from the human gene atlas project (available at https://www.pnas.org/content/101/16/6062 (accessed on 9 March 2020) from 158 expression experiments on healthy humans. From the input gene list, Expression Correlation computes a similarity matrix using expression levels from the reference database. Gene–gene correlation networks are then computed and visualized as an interaction network.

The biological function was related to the identified co-expressed gene clusters using the gene ontology (GO) database for molecular function. This allowed for the selection of specific gene-clusters that might pose as drug targets.

### 2.4. Biological Functional Enrichment

To perform biological functional enrichment, the R Bioconductor tool Cluster Profiler [30] was used to identify enriched GO, Kyoto Encyclopedia of Genes and Genomes (KEGG) pathways, and disease ontologies. Fisher’s exact test was performed to identify significantly enriched terms with a Benjamin-Hochberg adjusted probability value (BH adjusted *p*-value) of less than 0.05. This was conducted to relate biological function to the gene list related to HTN. Selecting genes as potential drug-targets with minimal influence on other biological pathways and processes was imperative to minimizing side effects in HTN patients.

### 2.5. Disease Ontology Enrichment

To identify and link the disease state to known HTN genes, disease ontology enrichment (DO) was performed using Cluster Profiler [30]. Disease ontology enrichment involved cross-referencing mappings to the Medical Subject Headings (MeSH) (available at https://www.nlm.nih.gov/databases (accessed on 9 March 2020), International Classification of Disease (ICD) (available at https://www.cdc.gov/nchs/icd/ (accessed on 9 March 2020), National Cancer Institute (NCI), (available at https://www.cancer.gov/research/resources (accessed on 9 March 2020), Systematized Nomenclature of Medicine-Clinical Terms (SNOMED CT) (available at http://www.snomed.org (accessed on 9 March 2020) and Online Mendelian Inheritance in Man (OMIM) (available at https://www.omim.org (accessed on 9 March 2020) databases. A hypergeometric model assessed the over-representation of the selected genes and their association with a disease. A BH-adjusted *p*-value was calculated to identify significantly associated diseases (BH *p*-value less than 0.05). This was performed to further identify possible HTN genes that show no association with other diseases. This was a possible indicator that the targeting chosen genes would have minimal interference with other biological functions.

## 3. Results

### 3.1. Selected Studies

We identified 2784 studies through data searches: PubMed (*n* = 410), Web of Science (*n* = 1115), Scopus (*n* = 1245), PharmGKB (*n* = 11), and African Journal Online (*n* = 3) (Figure 1). After removing duplicates (*n* = 1222) and studies that were conducted before molecular biology techniques (*n* = 43) the full text of 1519 publications were tested for suitability [28]. Of these, 1347 were excluded as follows; non-human studies (*n* = 29), non-HTN studies (*n* = 1194), non-African countries (*n* = 78), reviews (*n* = 46), and different study design (e.g., investigated the expression of genes in patients with HTN, clinical studies on HTN and family-based studies) (*n* = 130). Finally, 42 studies were included in this review.

The baseline characteristics of the included studies are summarized in Table 2. HTN was defined as systolic/diastolic BP (SBP/DBP) ≥ 140/90 mm Hg or the use of antihypertensive medications at inclusion [31,32]. Other studies used auscultatory DBP > 90 mm Hg or 24-h ambulatory DBP >85 mm Hg and auscultatory DBP > 95 mm Hg, SBP/DBP > 159/80 mm Hg, SBP/DBP >139/89 mm Hg and SBP/DBP ≥ 125/80 mm Hg, 25. This review did not identify a genome-wide association study (GWAS) conducted among the African population with HTN.

Northern African countries (*n* = 25) appeared to have more studies carried on HTN and SNPs association sub-Saharan African countries (Eastern Africa = 2, Central Africa = 1, Western Africa = 6, Southern Africa = 8) as elaborated in Figure 2. Twenty studies reported an association between HTN and genes such as *ACE, AGT, AGTR1, ANP, APOA5, ARGHGAP42, ATP2B1, B2, BAG6, CABCOCO1, CACNB2, CAND1, CHIC2, CNNM2, CPS1, CSK, CYP11B2, CYP2C8, EBF1, FES, FGF5, GNB3, GOSR2, GRK4, GUCY1A1N, HFE, IGFBP3, JAG1, LEP, MECOM, MOV10, MTHFR, NOS3, PLCE1, PLEKHA7, PR3, SH2B3, SLC39A8, SLC4A7, STK39, SUB1, TBX5, ULK4, ZNF652,* and *ZNF831* whereas twenty-two studies did not show any association (Table 2). The most studied genes were *AGT, ACE, NOS3, AGTR1, MTHFR, ATP2B1, CYP2C8, GNB3, CNNM2, PLEKHA7, JAG1, FGF5*, and *EBF1* as sequentially arranged from highest to the smallest number of studies (see Table 2). Other genes reported in single studies were *STK39, CDKAL1, IGF2BP2, TH, B2, CYP11B2, LEP, CLCNKB, SCNN1B, ADD1, ADRB2, SUB1, CEP83, IGFBP3, CHIC2, AGTR2, CPS1, MOV10, SLC4A7, MECOM, SLC39A8, GUCY1A1N, PR3, HFE, BAG6, CACNB2, PLCE1, CAND1, ARHGAP42, FES, GOSR2, ZNF831, ULK4, CABCOCO1, SH2B3, TBX5, CSK, and ZNF652*.

### 3.2. Gene–Drug Interaction and Ontology Analysis

A gene list consisting of 53 genes (Appendix A) linked to HTN was analyzed to gain an understanding of drug interaction, biological processes, pathways and various associated diseases Food and Drug Administration (FDA) approved drug list was obtained (Appendix A), and used to identify drug interactions with the above-mentioned genes. Identification of potential drug–gene interactions is the first step in a translation research effort aiming to reduce the burden of a major public health problem such as CVD [68]. Therefore, in the current study Drug Bank and CTD databases were utilized to generate a drug–gene interaction network (Figure 3) that identified the genes interacting with FDA approved HTN drugs. Of the 53 genes, only 14 genes (*CYP2C8, CYP11B2, AGT, AGTR1, AGTR2, ACE, ADRB2, LEP, MTHFR, NOS3, HFE, CNNM3, IGF2BP2*, and *SCNN1B*) showed an interaction with marketed drugs along with their corresponding side-effects (Figure 3). GO enrichment analysis was performed to identify the enriched biological processes that the gene list regulates (Figure 4). Finding ontologies linked to biological processes provides insights into the underlying mechanisms regulated by the chosen gene list. This allows for the identification of additional drugs that have previously not treated HTN.

Co-expression analysis was performed to identify co-expressed gene clusters (Figure 5). These were interrogated further to relate the molecular function for each gene cluster (Figure 6). This was performed to identify potential mechanisms in which drugs can be targeted against HTN. Genes in co-expressed gene clusters and their linked HTN drugs are summarized in Appendix A to explore potential combination therapy options.

### 3.3. Co-Expression Ontology Analysis

Similar to the GO enrichment analysis performed, KEGG pathway analysis (Figure 7) was performed to identify the enriched pathways regulated by the 53 HTN genes. This is performed to identify alternative pathways that interact with HTN related pathways, which can provide insights into how HTN drugs may influence various biological pathways.

### 3.4. Biological Pathway Analysis

Disease ontology (Figure 8) was also performed to analyze the involvement of the 53 HTN genes in various other diseases. This analysis gives insights into how different diseases may share common gene expression patterns and how drug treatment influences several biological pathways causing undesired side effects. In the selection process of identifying genes that are only linked to hypertension and other related diseases, DO analysis was conducted.

### 3.5. Disease Ontology

Disease ontology built from a disease-gene interaction database was plotted in a network (Figure 9) to visualize the interactions between genes and enriched diseases. This graph serves to identify genes that are targeted in HTN treatment however are also linked with other diseases.

## 4. Discussion

HTN is a multifactorial disease affecting one billion individuals. It is a leading cardiovascular risk factor accounting for premature deaths and has a significant economic cost [15]. Despite the availability of many antihypertensive drugs, less than 50% of patients have their blood pressure controlled [69]. These disappointing outcomes could be because of medication non-adherence and/or interindividual variation [4]. Considering studies linking interindividual variation to drug response, several studies have tested SNPs as predictors of TRH risk [70,71,72]. Various studies have reported an association between SNPs and HTN in European [73,74], African American [75], and Asian population [75,76,77], however not much evidence has been generated on the association of specific SNP or its associated haplotypes with HTN in an African population [27]. Therefore, we systematically extracted and discussed evidence on the African genetic variation and pharmacogenomics towards the treatment of HTN.

A total of 42 studies comprising of 53 genes are included in this review, however, only 20 studies reported a significant association with the risk of HTN. Table 2 highlighted candidate genes such as *ACE, NOS3, ATP2B1*, and *MTHFR* that have repeatedly been implicated to have an association with HTN in various African populations.

### 4.1. Angiotensin-Converting Enzyme Gene

Angiotensin-converting enzyme (*ACE*) is a metalloenzyme that cleaves angiotensin I to angiotensin II and inactivates a potent vasodilator [78]. *ACE* is encoded by the *ACE* gene which has been mapped to chromosome 17q23, and it has 26 exons and 25 introns [44]. Single-nucleotide polymorphism (rs1799752) on the *ACE* gene was investigated for its implication on high blood pressure in different sub-Saharan Africa countries (Table 2). It has been showed in various studies conducted in different parts of Africa continent that rs1799752 could be a potential genetic predictor for the development of HTN and permit initiation of personalized medicine [76,77,78]. However, other studies performed in South Africa, Tunisia, and Egypt exhibited no association of *ACE* gene polymorphism with the risk of developing HTN [38,40,79].

### 4.2. Nitric Oxide Synthase Gene

Nitric oxide synthase (*NOS3*) gene is another important candidate gene because of its critical role in regulating blood pressure. This gene (*NOS3*) is on chromosome 7q35–36, comprising 26 exons. *NOS3* is an integral component of vasorelaxing pathway, which is mediated by endothelium derived nitric oxide [80]. Various studies summarized in this review showed that different populations from Africa exhibit positive association of genetic variants of *NOS3* such as rs1799983 [47], rs2070744 [49], rs149868979 [58] and rs61722009 [51] with the risk of developing HTN. However, two studies in this review that aimed to investigate the *NOS3* gene variants associated with the development of HTN showed no association with the disease [48,50].

### 4.3. Plasma Membrane Calcium-Transporting ATPase 1 Gene

In 2009, the genome-wide association studies (GWAS) have identified the association between the Plasma membrane calcium-transporting ATPase 1 (*ATP2B1*) polymorphisms (rs2681472, rs17249754, rs2681492) and HTN in diverse populations [81,82]. This gene encodes for plasma membrane calcium dependent ATPase, which handles calcium pumping to the extracellular compartment [83]. In this review, it was revealed that genetic variants in *ATP2B1* genes in some populations from Algeria, Burkina Faso, and Uganda are associated with the risk of developing HTN. For instance, a study conducted in an Algerian population reported that the ATP2B1 variant (rs17249754) has a robust significant association with HTN [53]. The findings were further replicated in the Burkinabe population [3]. In another study conducted by Kayima et al. [33] on a Ugandan population, it was reported that rs2681492 was significantly associated with HTN in hypertensive individuals. Likewise, SNPs on *ATP2B1* (rs17249754, rs2681472, and rs2681492) were reported to be in linkage disequilibrium and located in the same linkage disequilibrium block in the Chinese population [84].

### 4.4. Methylenetetrahydrofolate Reductase Gene

Methylenetetrahydrofolate reductase gene (*MTHFR*) is one of the most studied genes associated with HTN in the African countries. *MTHFR* is on chromosome 1p36.3 and is well known to be involved in the metabolism of homocysteine and folate [85]. Previous studies have shown that the *MTHFR* variant such as rs1801133 has been associated with HTN between 24–87% [86,87,88]. Similarly, in this review, 50% of studies investigating the role of *MTHFR* provided a positive line of evidence linking this gene with HTN, showing that the rs1801133 polymorphism in *MTHFR* increases the risk of HTN [53,56]. Conversely, Amrani-Midoun et al. [47] and Amin et al. [55] in Algeria and Egypt showed no association of rs1801133 in hypertensive patients.

Furthermore, genes such as *CYP11B2* (rs1799998) [64], *LEP* (rs7799039) [65], and *CPS1* (rs1047891) [89] were also reported to be associated with HTN, however, these findings could not be replicated in the different African populations. As aforementioned, *ACE* and *NOS3* genes have been studied for its implication on HTN in more than six published papers indexed on the different databases, of which 50% of those were able to link these genes polymorphisms with the high risk for HTN. This suggests that there was a significant association of *ACE* and *NOS3* variants with HTN in African populations. However, *AGT, AGTR1*, and *AGTR2* polymorphisms in African populations showed less than 25% or no association with HTN. For example, all the SNPs located near *AGT* gene, only rs2004776 was demonstrated to have a strong association with HTN in Ugandan population [81]. In a recent study by EI-Garawani et al. [90], two polymorphisms (rs4762 and rs699) of *AGT* gene were found to be associated with the risk of diabetes and hypertension in the Egyptian population. Furthermore, Farrag et al. [42] reported the positive association of *AGTR1*, rs5186 with HTN in Egyptian population and similar findings were also observed in the Ghanaian population. However, no association has been recorded in other *AGTR1* and *AGTR2* polymorphisms.

### 4.5. Computational Insights

Furthermore, we attempted to shortlist key target genes out of the 53 identified genes in this review by using bioinformatics methods on gene-specific data available in various public databases. The bioinformatics analysis investigated various parameters associated with the 53 HTN genes such as drug interactions, co-expression of genes and disease ontology, and linked pathways. Only 14 genes (*CYP2C8, CYP11B2, AGT, AGTR1, AGTR2, ACE, ADRB2, LEP, MTHFR, NOS3, HFE, CNNM3, IGF2BP2*, and *SCNN1B*) were identified to have an interaction with FDA approved marketed HTN drugs (Figure 3 and Appendix A). As evident from Figure 3, the various antihypertensive drugs such as losartan, acebutolol, atenolol, metoprolol, ramipril, lisinopril, labetalol, and amlodipine show distinct side-effects. It has also been reported in the literature [91,92,93] that 20–97% of patients taking antihypertensive drugs suffer from various drug-related side effects. Furthermore, these side effects may potentially also account for non-adherence to antihypertensive drugs [94,95,96]. Losartan, an angiotensin II antagonist and atenolol, a **β**-blocker showed the most side-effects related to respiratory impairments and infection. Prior studies have also documented a higher frequency of side-effects and lower adherence among patients taking diuretics and **β**-blockers [97]. In hypertensive patients of African ancestry, diuretics and or calcium channel blockers (amlodipine) are preferably recommended [98] and patients who remain concerned about the adverse health effects of antihypertensive drugs are less likely to be adherent to their medications [99].

The co-expression analysis identified three clusters, including cluster 1 (*ACE, AGT, AGTR1, AGTR2,* and *NOS3), cluster 2 (MOV10, CAN1*, and *IGF2BP2*), and cluster 3 (*CSK* and *ADRB2*). Upon further analysis, from cluster 1, *AGTR1, AGTR2, AGT*, and *ACE*; cluster 2, *IGFBP2* and cluster 3 *ADRB2* genes are shown to be mapped to HTN drugs shown in Figure 3 and Appendix A. Interestingly, the protein-coding genes *AGTR1, AGTR2, AGT*, and *ACE* are all involved in the RAAS cascade which regulates blood pressure and vascular resistance. As such, the most widely used HTN drug classes include *ACE* inhibitors, angiotensin II antagonists and direct renin inhibitors which target the RAAS cascade [100]. Whereas, in the second and third cluster, the genes *IGFBP2* (targeted by vasodilators) and *ADRB2* (targeted by **β**-blockers), respectively, have been recently linked with increased risk of HTN and have been recently proposed as potential diagnostic biomarkers for HTN. For example, in a study by Yang et al. [101] the overexpression of the gene *IGFBP2* which is linked to the regulation of vascularization and angiogenesis is associated with the increased severity of HTN. Additionally, the gene *ADRB2* has been shown to interact with the gene *NOS3* [102]. *ADRB2* may indirectly interact with the RAAS cascade via the *NOS3* forming part of the gene cluster of genes associated with the RAAS cascade (Figure 5). This is further illustrated in Figure 7 where *ADRB2* shows interaction with RAAS associated genes by regulating common biological pathways including renin secretion and adrenergic signaling. Lastly, the observations in Figure 7, Figure 8 and Figure 9 highlight the genes *ADRB2*, *ACE, AGT*, and *NOS3* to have an association with the metabolic syndrome (includes hypertension, type II diabetes, excess lipids and abnormal cholesterol levels). Although further validation is required, the above-mentioned genes exhibit characteristics of being key regulators in metabolic syndrome.

This is also evident from the biological processes and pathways showed enrichment of HTN (by vasoconstriction and RAAS) and diabetic-related terms (see Figure 6 and Figure 7). *NOS3* (mapped to drug–gene interaction network) and PLCE1 showed no interaction with HTN related KEGG pathways but exclusive to AGE-RAGE signaling pathways in diabetic complications. As such, using drugs targeted against HTN may influence other signaling pathways. In this case, *NOS3* interacts with 3 HTN drugs (Angiotensin II antagonists, calcium channel blockers, and β-blockers).

It is also important to understand the roles that genes play in different sub-types of HTN. Figure 8 shows that certain genes are shared among different sub-types of HTN and there are genes which contribute to a specific sub-type of the disease. For example, *CLCNKB* is only linked to *TRH*, therefore, a drug targeting this gene will be effective in treating salt sensitive HTN, since a polymorphism in this gene leads to increased salt retention in the bloodstream. Similarly, the gene such as *ACE* is linked to almost all categories of diseases shown in Figure 8, thus pointing towards its universal role in HTN related diseases.

From the DO analysis, the following potential gene targets have been selected, these include *CLCNKB, CYPB11B2, SH2B2, STK9*, and *TBX5* as they show interactions exclusively with hypertension-related pathways (see Figure 8). Furthermore, targeting co-regulated gene clusters as opposed to single-gene targeting increases the likelihood of more effective drug treatment for HTN. Cluster 1 (*ACE, AGT, AGTR1, AGTR2*, and *NOS3*) and cluster 3 (*CSK* and *ADRG1*) showed enrichment of G-coupled receptor binding activity (a classical target of drug discovery) and protein tyrosine kinase receptors (see Figure 5), thus demonstrates that these gene clusters can be targeted during future drug design. Gene–gene interactions among regulatory variants play a critical role in the drug–response phenotype [103,104]. Based on the previous research, and findings in the current research paper it has been suggested that all genes within a cluster should be screened and molecular mechanisms supporting various gene–gene interactions should be explored that may enable identification of potential pharmacogenomic biomarkers to guide personalized medicine approaches for HTN. Therefore, we anticipate that these potential genes including their SNPs need to be investigated further in African populations.

While several studies in this review support an important contribution of genetic polymorphisms in HTN, some studies have failed to detect significant effects or replicate previous findings. These inconsistencies may be linked to the African population’s high genetic diversity and low levels of linkage disequilibrium among loci when compared to populations from other countries outside Africa [105]. Furthermore, studies suggest that from a genetic standpoint, there is no SNP-database that can be used to personalize treatments for HTN in an African population [47,106,107]. Inferring that the lack of genetic information with robust allele frequency distributions serves as a hurdle to implement corrective treatment and this may have important medical implications [108]. Providing a more accurate reference foundation on which to support future disease research in Africa is thus of utmost importance [27].

## 5. Conclusions

In conclusion, the present study reported on the susceptibility of inter-individual genetic variation to HTN in African population. Large-scale genetic studies are needed to better understand the susceptibility of African population-based inter-individual genetic variation and their effect on the hypertensive drug response, which will aid in the development of effective African based individualized antihypertensive medicine.

## Figures and Tables

**Figure 1 genes-12-00532-f001:**
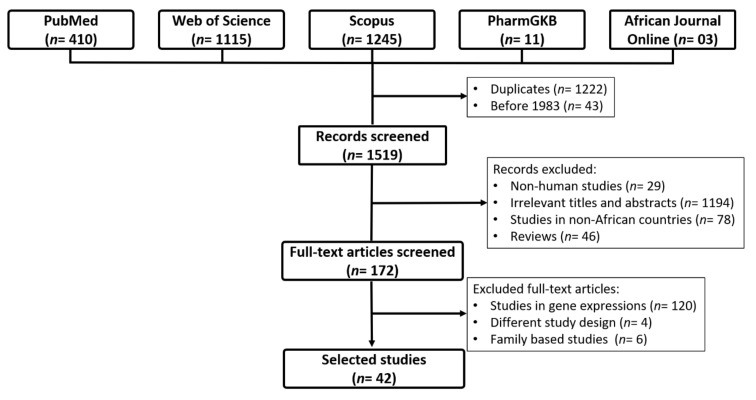
Flowchart for the study selection.

**Figure 2 genes-12-00532-f002:**
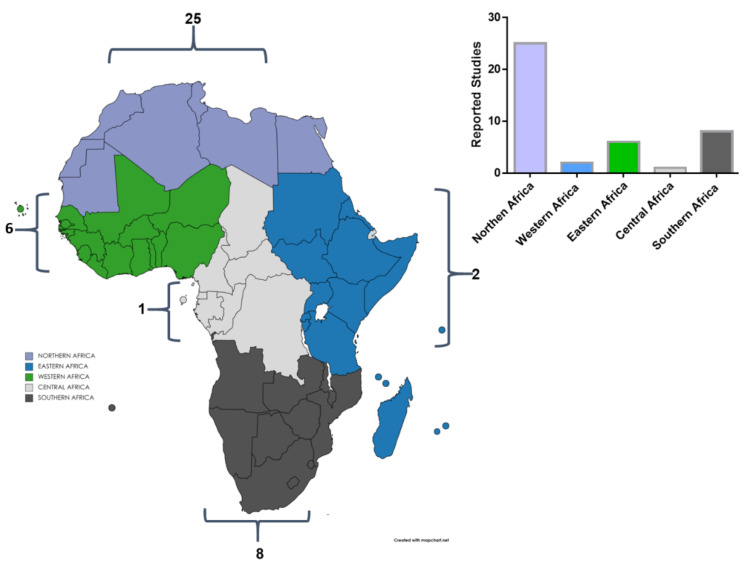
Summary of all genetic studies reported in the African continent in relation to hypertension.

**Figure 3 genes-12-00532-f003:**
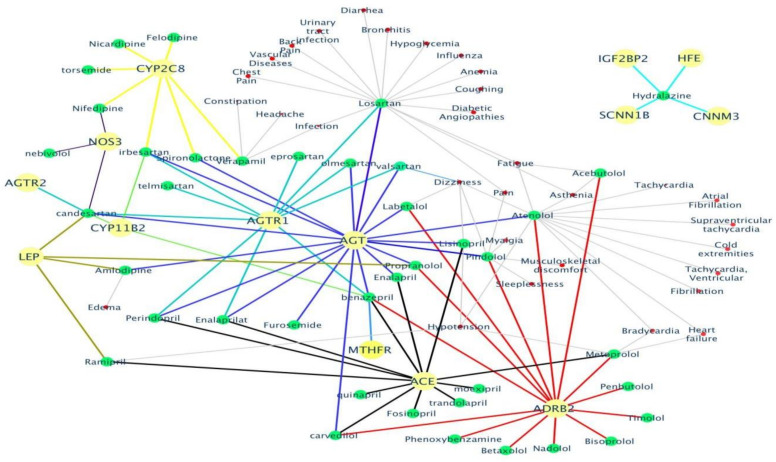
The identified drug–gene interactions along with the side-effects are plotted into a network. Only 14 out of the total 53 HTN genes (shown as yellow nodes) mapped to 57 FDA approved HTN drugs (shown as green nodes) and their corresponding side-effects (shown as red nodes). The edges in the drug–gene interaction network are shown in a different color for each gene to easily distinguish their association with the various HTN drugs.

**Figure 4 genes-12-00532-f004:**
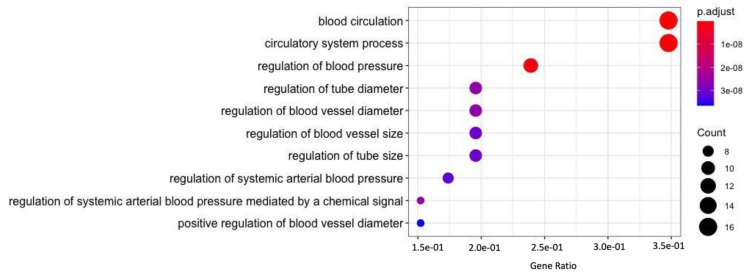
The gene ontology analysis performed on all 53 prioritized HTN related genes to identify related biological processes. The gene ratio of participating genes in the enriched ontology, the color-coded Benjamini-Hochberg (BH)-adjusted *p*-value, and the number of genes (count) in each enriched ontology are shown in the above plot.

**Figure 5 genes-12-00532-f005:**
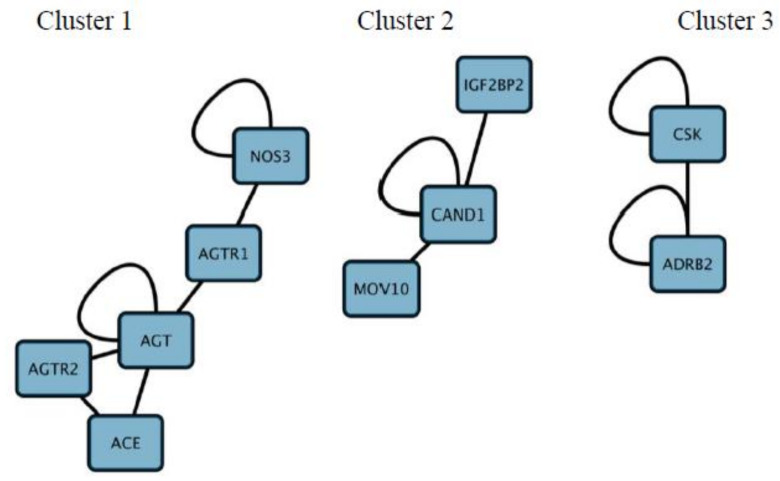
Co-expression networks generated to identify co-regulated gene clusters from the 53 prioritized genes related to Hypertension Co-expressed gene clusters have been numbered above from 1 to 3.

**Figure 6 genes-12-00532-f006:**
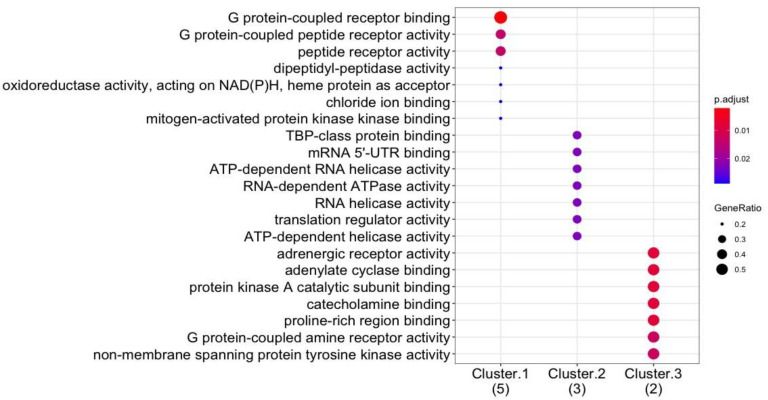
GO enrichment (i.e., molecular function) performed on the co-expressed gene clusters (labelled cluster 1–3, and the number of genes in brackets) to relate molecular function to co-expressed genes. The gene ratio of participating genes in the enriched ontology, and the color coded BH-adjusted *p*-value are shown in the plot.

**Figure 7 genes-12-00532-f007:**
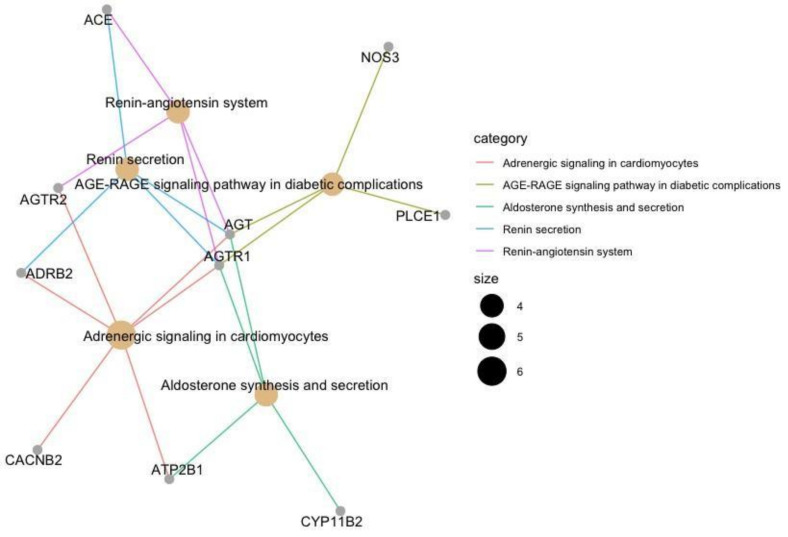
Pathway enrichment analysis performed on the 53 HTN genes annotated using the KEGG database. Only 10 from the 53 HTN genes mapped to the KEGG database linking them to biological pathways. The plot illustrates the participating genes with the enriched pathway. Each enriched pathway has been color-coded, and the number of participating genes corresponds to the size of the node.

**Figure 8 genes-12-00532-f008:**
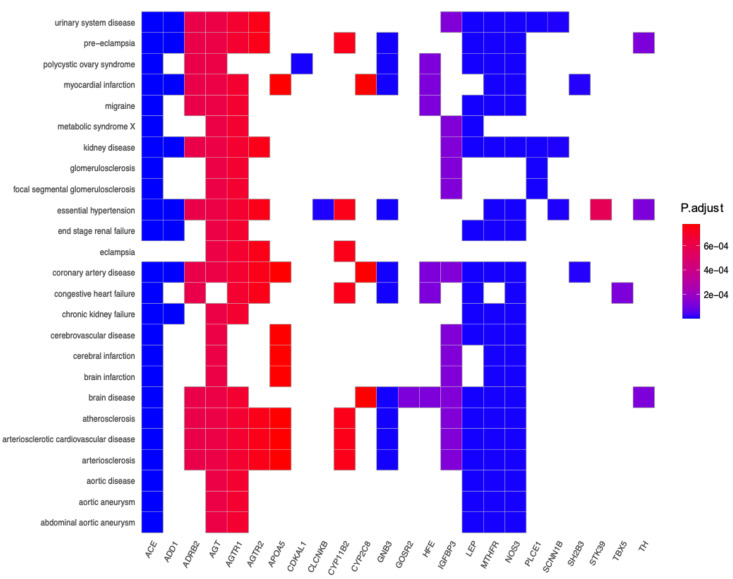
The enrichment of disease ontology for the mapped genes related to hypertension. Each gene is statistically mapped to a disease. The matched boxes associated with the enriched disease term are color coded by the BH-adjusted *p*-values.

**Figure 9 genes-12-00532-f009:**
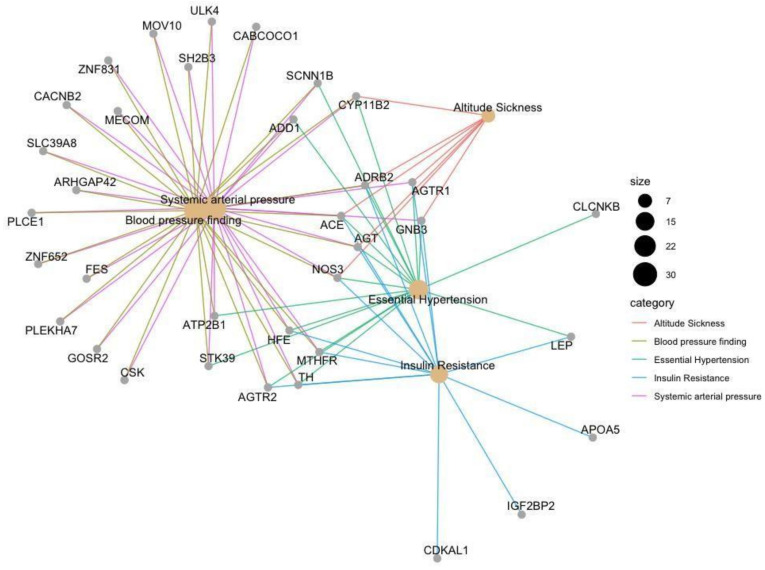
The gene–disease interaction network of the mapped 53 genes related to HTN to the disease ontology database. The size of the enriched disease illustrated as a node corresponds to the number of participating genes. The color of the edges corresponds to the enriched disease.

**Table 1 genes-12-00532-t001:** Inclusion Criteria and Data Extraction.

Inclusion	Exclusion
African population	Studies in non-African countries
Published from 1984 to 2020	Studies conducted before 1983
Human studies	Non-human studies
Studies investigating an association (*p* < 0.05) between SNP and hypertension.	Studies in gene expression
Case-control design	Reviews
	Family-based studies

**Table 2 genes-12-00532-t002:** Hypertension and single-nucleotide polymorphisms (SNPs) association in Africa.

Gene	Chr Position	SNP	Alleles	Alt Allele Freq, Global (db. SNP)	Alt Allele Freq, African (db. SNP)	Cases No.	Controls	Association (*p* < 0.05)	Country	Author Year
*AGT*	1q42.2	rs2004776	C > G	0.410	0.487	782	2099	Yes	Uganda	Kayima et al., 2017 [33]
		rs4762	G > A	0.102	0.054	75	70	No	Algeria	Amrani et al., 2015 [34]
		rs699	A > G	0.705	0.903	202	204	No	Burkina Faso	Tchelougou et al., 2015 [35]
						612	612	No	Nigeria	Kooffreh et al., 2014 [36]
						81	178	No	Algeria	Meroufel et al., 2014 [37]
						110	93	No	Egypt	AbdRaboh et al. 2012 [38]
						39	22	No	Tunisia	ALrefai et al., 2010 [39]
						195	107	No	South African	Ranjith et al., 2004 [40]
*AGTR1*	3q24	rs5186	A > C	0.118	0.020	36	50	No	Cameroon	Ghogomu et al., 2016 [41]
						202	204	No	Burkina Faso	Tchelougou et al., 2015 [35]
						81	178	No	Algeria	Meroufel et al., 2014 [37]
						612	612	No	Nigeria	Kooffreh et al., 2014 [36]
						142	191	No	Tunisia	Mehri et al., 2012 [38]
						40	15	Yes	Egypt	Farrag et al., 2011 [42]
						195	107	No	South African	Ranjith et al., 2004 [40]
						NA	NA	Yes	Ghana	Williams et al., 2004 [43]
*ACE*	17q23.3	rs1799752	NA	NA	NA	202	204	Yes	Burkina Faso	Tchelougou et al., 2015 [35]
						217	161	Yes	Egypt	Zawilla et al., 2014 [44]
						110	93	No	Egypt	AbdRaboh et al. 2012
						40	21	Yes	Egypt	Badr et al., 2012 [45]
						142	191	No	Tunisia	Mehri et al., 2012 [38]
						40	40	Yes	Egypt	Bessa et al., 2009 [46]
						195	107	No	South African	Ranjith et al., 2004 [40]
*NOS3*	7q36.1	rs1799983	T > A	0.824	0.930	77	77	Yes	Algeria	Amrani-Midoun et al., 2019 [47]
						145	184	Yes	Morocco	Nassereddine et al., 2018 [25]
		rs2070744 intron 4 VNTR	C > T N/A	0.766 N/A	0.862 N/A	157	144	No No Yes	Sudan	Gamil et al., 2017 [48]
		rs1799983	T > A	0.824	0.930	70	30	Yes	Tunisia	ALrefai et al., 2016 [39]
		rs2070744	C > T	0.766	0.862	288	373	Yes	Tunisia	Jemaa et al., 2011 [49]
		rs1799983	NA	NA	NA	537	565	No	Tunisia	Sediri et al., 2010 [50]
		rs61722009	NA	NA	NA	295	395	Yes	Tunisia	Jemaa et al., 2009 [51]
*MTHFR*	1p36.3	rs1801133	G > A	0.245	0.090	82	72	No	Algeria	Amrani-Midoun et al., 2016 [52]
						189	598	Yes	Algeria	Lardjam-Hetraf et al.,2015 [53]
						101	102	Yes	Morocco	Nassereddine et al., 2015 [54]
						97	84	No	Egypt	Amin et al., 2012 [55]
*ATP2B1*	12q21.q23	rs2681472 rs17249754	A > G G > A	0.199 0.209	0.094 0.131	180	200	Yes Yes	Burkina Faso	Sombie et al., 2019 [3]
		rs2681492	T > C	0.208	0.126	782	2099	Yes	Uganda	Kayima et al., 2017 [33]
		rs17249754	G > A	0.209	0.131	189	598	Yes	Algeria	Lardjam-Hetraf et al.,2015 [53]
*CLCNKB*	1p36.3	rs12140311	A > T	0.098	0.214	213	545	Yes	Ghana	Sile et al., 2009 [56]
		rs34561376	G > A	0.082	0.142	213	545	No	Ghana	Sile et al., 2007 [57]
*GNB3*	12p13.31	rs5443	NA	NA	NA	388	425	No	Tunisia	Kabadou et al., 2013 [31]
		rs74837985	NA	NA	NA	40	40	Yes	Egypt	Bessa et al., 2009 [46]
*CNNM2*	10q24.32	rs11191548	T > C	0.152	0.025	782	2099	Yes	Uganda	Kayima et al., 2017 [33]
						189	598	Yes	Algeria	Lardjam-Hetraf et al.,2015 [53]
*PLEKHA7*	11p15.2	rs381815	C > A	0.206	0.190	782	2099	Yes	Uganda	Kayima et al., 2017 [33]
						189	598	Yes	Algeria	Lardjam-Hetraf et al.,2015 [53]
*JAG1*	20p12.2	rs1327235	A > G	0.464	0.494	782	2099	Yes	Uganda	Kayima et al., 2017 [33]
						189	598	Yes	Algeria	Lardjam-Hetraf et al.,2015 [53]
*SCNN1B*	16p12.2	rs149868979	NA	NA	NA	1468	471	Yes	South Africa	Jones et al., 2012 [58]
		rs1799979	C > T	0.007	0.024	519	514	No	South Africa	Nkeh et al., 2003 [59]
*FGF5*	4q21.21	rs1458038	C > T	0.230	0.037	782	2099	Yes	Uganda	Kayima et al., 2017 [33]
						189	598	Yes	Algeria	Lardjam-Hetraf et al.,2015 [53]
*EBF1*	5q33.3	rs11953630	C > A	0.07	0.180	782	2099	Yes	Uganda	Kayima et al., 2017 [33]
						189	598	Yes	Algeria	Lardjam-Hetraf et al.,2015 [53]
*STK39*	2q24.3	rs3754777	C > T	0.195	0.119	180	200	Yes	Burkina Faso	Sombie et al., 2019 [3]
*APOA5*	11q23.3	rs662799 rs3135506 rs2075291	G > A G > A C > A	0.837 0.056 0.011	0.884 0.067 0.002	149	134	Yes	Morocco	Ouatou et al., 2014 [60]
*CDKAL1* *IGF2BP2*	6p22.3 3q27.2	rs7756992 rs4402960	A > G G > T	0.413 0.389	0.633 0.567	200	208	No	Tunisia	Lasram et al., 2015 [61]
*TH*	11p15.5	C824T	NA	NA	NA	200	202	No	South Africa	van Deventer et al., 2013 [62]
*B2*	14q32.1-q32.2	B_2_ C-58T B_2_ -9/+9	NA NA	NA NA	NA NA	88	77	Yes	South Africa	Moholisa et al., 2013 [63]
*CYP11B2*	8q24.3	rs1799998	A > G	0.347	0.189	537	565	Yes	Tunisia	Saidi et al., 2010 [64]
*NPPA*	1p12	rs748566461 C1364A C55A	NA	NA	NA	298	278	Yes	South Africa	Nkeh et al., 2002 [32]
*CYP2C8*	10q23.331	rs10509681 rs11572080	T > C C > T	0.046 0.046	0.008 0.008	NA	NA	Yes	Ghana	Williams et al., 2004 [43]
*LEP*	7q32.1	rs7799039	G > A	0.402	0.032	45	53	Yes	Tunisia	Ben et al., 2008 [65]
*ADD1*	4p16.3	rs4961	G > T	0.208	0.049	148	94	No	South Africa	Barlassina et al., 2000 [66]
*ADRB2*	5q32	rs1042713 rs1042714	G > A G > C	0.476 0.796	0.520 0.864	192	123	No	South Africa	Candy et al., 2000 [66]
*SUB1* *CEP83* *IGFBP3* *CHIC2*	5p13.3 12q22 7p12.3 4q12	rs7726475 rs11837544 rs11977526 rs11725861	G > A T > A G > A A > G	0.187 0.081 0.442 0.169	0.024 0.216 0.326 0.181	782	2099	Yes	Uganda	Kayima et al., 2017 [33]
*AGTR2*	Xq23	rs11091046	NA	NA	NA	382	403	No	Tunisia	Kabadou et al., 2012 [67]
*CPS1*	2q34	rs1047891	C > A	0.289	0.368	NA	NA	Yes	Ghana	Williams et al., 2004 [43]
*MOV10* *SLC4A7* *MECOM* *SLC39A8* *GUCY1A1N* *PR3* *HFE* *BAG6* *CACNB2* *PLCE1* *CAND1* *ARHGAP42* *FES* *GOSR2* *ZNF831* *ULK4* *CABCOCO1* *SH2B3* *TBX5* *CSK* *ZNF652*	1p13.2 3p24.1 3q26.2 4q24 4q32.1 19p13.3 6p22.2 6p21.33 10p12.33 10q23.33 12q14 11q22.1 15q26.1 17q21.32 20q13.32 3p22.1 10q21.2 12q24.12 12q24.21 15q24.1 17q21.32	rs2932538 rs13082711 rs419076 rs13107325 rs13139571 rs1173771 rs1799945 rs805303 rs4373814 rs932764 rs7129220 rs633185 rs2521501 rs17608766 rs6015450 rs3774372 rs4590817 rs3184504 rs10850411 rs1378942 rs12940887	A > C T > C T > A C > A C > A A > G C > G G > A G > C A > G G > A G > A A > C T > C A > G T > C G > C T > A T > C C > A C > T	0.830 0.120 0.584 0.024 0.211 0.661 0.073 0.436 0.512 0.428 0.058 0.639 0.213 0.054 0.098 0.173 0.110 0.853 0.470 0.245 0.185	0.842 0.036 0.450 0.002 0.129 0.795 0.011 0.643 0.613 0.184 0.052 0.804 0.223 0.010 0.205 0.197 0.190 0.981 0.346 0.026 0.048	189	598	Yes	Algeria	Lardjam-Hetraf et al., 2015 [53]

## Data Availability

Not applicable.

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
