# Peer review of "Hypertension in African Populations: Review and Computational Insights"

_genes, 2021, doi:10.3390/genes12040532_

Round 1

Reviewer 1 Report

This interesting systematic review by Sihle et al discussed the African evidence on the genetic variation and pharmacogenomics towards the treatment of hypertension. Further, in silico methods are utilized to elucidate biological processes that will aid in identifying novel drug targets for the treatment of UHTN in an African population.

This review was designed and written well. The following minor concerns are raised to improve clarity

  1. Please define the UHTN in the abstract.
  2. Did authors analyze the data based on sex, gender, and age to determine the genetic variations in the African population, If so it can be included in their review?
  3. Along with the genetic polymorphism, there is a cluster of gene interactions associated with hypertension, what kind of strategies can apply to treat such individuals needed to discuss.

Reviewer 2 Report

Mabhida et al present a systemic review of hypertension in African populations.  This is a great review with the potential to reach a wide spectrum of research interests.  It is well-written and very informative.  The authors have gone beyond the traditional review approach by conducting in silico analysis to predict potential drug targets, drug-gene interactions, biological processes, co-expressed genes, regulatory pathways, and disease-gene interactions.

Concerns/comments:

  1. What is the significance threshold used for the inclusion of genetic associations? P<0.05? Should be added to Table 1.
  2. The term in silico should be in italics, in silico.
  3. UHTN is not defined, uncontrolled hypertensive (UHTN).
  4. Single-nucleotide polymorphism is not hyphenated.
  5. Need indentation on paragraph on page 3, line 95.
  6. On page 19 line 387, the phrase “however not much evidence has been got” should be changed to: not much evidence has been generated/produced/obtained; or not much evidence exists on…Been got should not be used.
